# RAD-TTS: Parallel Flow-Based TTS with Robust Alignment Learning and Diverse Synthesis

**Kevin Shih** [1]  **Rafael Valle** [1]  **Rohan Badlani** [1]  **Adrian Łańcucki** [1]  **Wei Ping** [1]  **Bryan Catanzaro** [1]

## Abstract

This work introduces a predominantly parallel, end-to-end TTS model based on normalizing flows. It extends prior parallel approaches by additionally modeling speech rhythm as a separate generative distribution to facilitate variable token duration during inference. We further propose a robust framework for the on-line extraction of speech-text alignments – a critical yet highly unstable learning problem in end-to-end TTS frameworks. Our experiments demonstrate that our proposed techniques yield improved alignment quality, better output diversity compared to controlled baselines.

## 1. Introduction

While speech synthesis is most naturally modeled sequentially in a fully autoregressive manner (Wang et al., 2017; Valle et al., 2020), training and inference speeds scale poorly with the sequence length. Furthermore, a single poorly predicted audio frame in a fully autoregressive model can catastrophically impact all subsequent inference steps, prohibiting its use for inferring long sequences. Recent works (Kim et al., 2020; Miao et al., 2020; Peng et al., 2020) have presented increasingly parallel solutions to address these concern. They first determine the duration of each phoneme in the input text, then use a generative, parallel architecture to sample and decode each mel-spectrogram frame in parallel rather than sequentially. However, parallel architectures present challenges of their own. The following work proposes RAD-TTS: a text-to-speech (TTS) framework featuring robust alignment learning and diverse synthesis. We propose a stable and unsupervised alignment-learning framework applicable to virtually any TTS framework, as well as a generative phoneme-duration model to ensure diverse output in parallel TTS architectures.

---

[*]Equal contribution  [1]NVIDIA. Correspondence to: Kevin Shih <kshih@nvidia.com>.

Third workshop on *Invertible Neural Networks, Normalizing Flows, and Explicit Likelihood Models* (ICML 2021). Copyright 2021 by the author(s).

Learning unsupervised audio-text alignment is difficult, especially in parallel architectures. Some works use off-the-shelf forced aligners (Ren et al., 2020), while other opt to distill the attention from an autoregressive model (or forced aligner) into a parallel architecture in an expensive two-stage process (Peng et al., 2020; Ren et al., 2019; Łańcucki, 2020), which makes the training even more expensive. Obtaining alignments from an external aligner has severe limitations, as it requires finding or training an external aligner for every language and alphabet and requires near-perfect text normalization. Most related to our work is Glow-TTS (Kim et al., 2020), which proposes an aligning mechanism in a normalizing flow framework. Our work extends their alignment setup to be more generally applicable to arbitrary TTS frameworks, as well as with improved stability.

We further tackle the limited synthesis diversity in parallel TTS architectures. These architectures first determine the durations of each phoneme. Then, a parallel architecture maps the phonemes, replicated time-wise based on their predicted durations, to corresponding Mel-spectrogram frames. Even if the latter parallel architecture is generative, the phoneme prediction is typically deterministic. This limits diversity in output as a lot of the *variability is dependent on speech rhythm*. Even flow-based ones such as Glow-TTS use deterministic regression models to infer the duration. As such, we propose to use a separate generative model only for token durations. Our results demonstrate more diverse inference results in comparison to a fixed-duration baseline.

In summary, our work proposes 1) a rapidly converging alignment learning framework incorporating the forward algorithm and priors, and 2) Generative duration modeling to ensure diverse samples in parallel TTS architectures.

## 2. Method

We aim to construct a generative model for sampling mel-spectrograms given text and speaker information. We extend prior bipartite-flow TTS methods (Kim et al., 2020; Miao et al., 2020) with a more robust alignment learning mechanism (2.2.1) and generative duration modeling (2.3). We formulate our setup as follows: consider an audio clip of human speech represented as a mel-spectrogram tensor

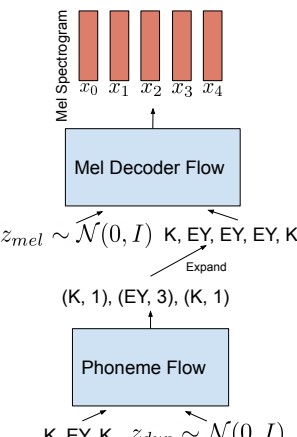

*Figure 1.* Simplified inference pipeline for the proposed model. The phoneme flow first samples from $P_{dur}()$ to attain per-phoneme durations $(\mathcal{A})$, which are then used to prepare the input to the parallel Mel-Decoder flow that models $P_{mel}()$

$X \in \mathbb{R}^{C_{mel} \times T}$, where $T$ is the number of mel-frames over the temporal axis, and $C_{mel}$ is the number of bands or dimensions per frame. Next, let $\Phi \in \mathbb{R}^{C_{txt} \times N}$ be a tensor of embedded text sequence of length $N$, and $\mathcal{A} \in \mathbb{R}^{N \times T}$ be a matrix of temporal alignment between the audio clip and the text (see Fig. 2). Finally, let $\xi$ be a vector that encodes speaker-specific characteristics. We model the conditional distribution:

$$P(X, \mathcal{A}|\Phi, \xi) = P_{mel}(X|\Phi, \xi, \mathcal{A})P_{dur}(\mathcal{A}|\Phi, \xi) \quad (1)$$

such that we can sample *both* mel-spectrogram frames and their durations at inference time while maintaining a parallel architecture for modeling $P_{mel}$. An overview of this pipeline is shown in Fig. 1.

### 2.1. Normalizing Flows

We begin with a overview of normalizing flows as applied to mel-decoding in TTS. Let $p_X(x)$ represent the unknown likelihood function for each mel-frame ($P_{mel}()$ with conditionals omitted for brevity). We wish to model the distribution such that each time step $x \in X$ can be sampled i.i.d. from a standard Normal. To achieve this, we fit an invertible function $g$ such that $z = g^{-1}(x)$ and define $z$ such that $z \sim \mathcal{N}(0, \mathbf{I})$. Using the change of variables function:

$$p_X(x) = p_Z(z) \left| \det \mathbf{J}(g(z)) \right|^{-1}, \quad (2)$$

where $\mathbf{J}$ is the Jacobian of the invertible transformation, and $p_Z(z)$ is the Gaussian likelihood function $\mathcal{N}(z; 0, \mathbf{I})$. Our resulting maximum log-likelihood objective with respect to data samples $x$ is written as:

$$\log p_X(x) = \log p_Z(g^{-1}(x)) + \log \left| \det \mathbf{J}(g^{-1}(x)) \right|, \quad (3)$$

where we achieve exact MLE by finding the parameters for $g$ that maximize the right-hand-side. Inference is performed by: $z \sim \mathcal{N}(0, \mathbf{I}) \quad$ and $\quad x = g(z)$.

### 2.2. Mel Decoder Architecture

Our mel-decoder model, which we will continue to denote as $g$, models only $P_{mel}()$ in Eq. 1, though a similar formulation is used to model $P_{dur}()$ as we will show later on. The decoder allows us to sample latent vectors $z$ for each time step i.i.d. from a Gaussian prior, and map them to plausible-sounding mel-frames $x$. This transformation needs to be invertible to satisfy the change of variables requirement, and its behavior needs to be conditioned on text $\Phi$, speaker $\xi$, and alignment $\mathcal{A}$ information. Following prior works (Kingma & Dhariwal, 2018; Dinh et al., 2014; 2016), we construct $g$ as a composition of invertible functions, specifically:

$$X = g(Z; \Phi, \xi, \mathcal{A}) = g_1 \circ g_2 \ldots g_{K-1} \circ g_K(Z; \Phi, \xi, \mathcal{A}) \quad (4)$$

$$Z = g_K^{-1} \circ g_{K-1}^{-1} \ldots g_2^{-1} \circ g_1^{-1}(X; \Phi, \xi, \mathcal{A}) \quad (5)$$

Note that $X$ is the entire mel-spectrogram sequence for any audio clip and $Z$ is its corresponding projection into the latent domain with the same dimensions as $X$, and $z \sim \mathcal{N}(0, \mathbf{I})$ for each $z \in Z$. Each compositional unit $g_k(X; \Phi, \xi, \mathcal{A})$ is an invertible network layer of $g$, henceforth referred to as a step of flow.

As with previous works (Kim et al., 2020; Miao et al., 2020; Prenger et al., 2019), we use a Glow-based (Kingma & Dhariwal, 2018) bipartite-flow architecture, where each step of flow is a $1 \times 1$ invertible convolution (Kingma & Dhariwal, 2018) paired with an affine coupling layer (Dinh et al., 2016). Detailed descriptions provided in the Appendix.

#### 2.2.1. UNSUPERVISED ALIGNMENT LEARNING

We devise an unsupervised learning approach that learns the alignment between text and speech rapidly without depending on external aligners. The alignment converges rapidly towards a usable state in a few thousand iterations (roughly ten-twenty minutes into training). In order to learn to align $X$ with $\Phi$, we combine the Viterbi and forward-backward algorithms used in Hidden Markov Models (HMMs) (Rabiner, 1990) in order to learn hard and soft alignments, respectively. Let $\mathcal{A}_{soft} \in \mathbb{R}^{N \times T}$ be the alignment matrix between text $\Phi$ and mel-frames $X$ of lengths $N$ and $T$ respectively, such that every column of $\mathcal{A}_{soft}$ is normalized to a probability distribution. Sample alignment matrices are shown in Figure 2. Our goal is to extract monotonic, binarized alignment matrices $\mathcal{A}_{hard}$ such that for every frame the probability mass is concentrated on a single symbol, and $\sum_{j=1}^{T} \mathcal{A}_{hard,i,j}$ yields a vector of durations of every symbol.

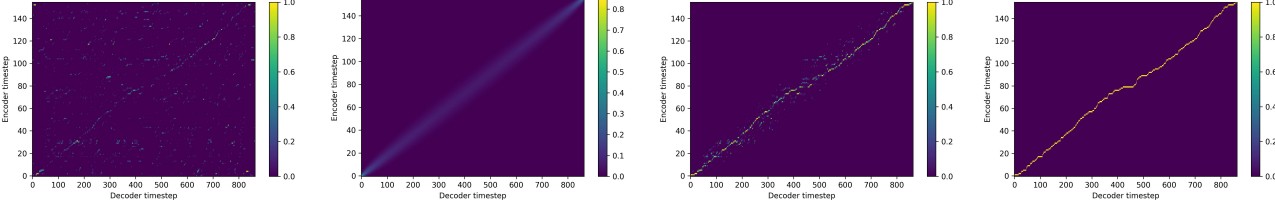

(a) Soft alignment $\mathcal{A}_{soft}$      (b) Beta-binomial Prior      (c) $\mathcal{A}_{soft}$ w/ Prior      (d) Hard alignment $\mathcal{A}_{hard}$

*Figure 2.* Visualizations of the alignment attention matrices $\mathcal{A}$. The vertical axis represents text tokens from bottom to top. The horizontal axis represents mel-frames from left to right. Fig. 2a shows the baseline soft attention map. Fig. 2c combines Fig. 2a with Fig. 2b to penalize alignments straying too far from the diagonal. Fig. 2d is the most likely monotonic alignment extracted from Fig. 2c with Viterbi.

**Extracting $\mathcal{A}_{soft}$:** Similar to Glow-TTS, the soft alignment distribution is based on the learned pairwise affinity between all text tokens $\phi \in \Phi$ and mel-frames $x \in X$, which is then normalized with softmax across the text dimension:

$$D_{i,j} = dist_{L2}(\phi_i^{enc}, x_j^{enc}), \qquad (6)$$

$$\mathcal{A}_{soft} = \texttt{softmax}(-D, \texttt{dim}=0). \qquad (7)$$

Here, $x^{enc}$ and $\phi^{enc}$ are encoded variants of $x$ and $\phi$, each using a 2-3 layer 1D CNNs. We found that simple transformations with limited receptive fields produced best results. Conversely, expensive, wide receptive field transformations resulted in instabilities.

The forward-backwards algorithm maximizes the likelihood of hidden states given observations. We consider only with the forward probabilities. Text is defined as the hidden state and the mel as the observation to maximize $P(S(\Phi)|X)$, considering all valid monotonic assignments to sequences $\mathbf{s} \in S(\Phi)$ where a specific sequence might look like: $\mathbf{s} : \{s_1 = \phi_1, s_2 = \phi_2, \dots s_T = \phi_N\}$. A **monotonic sequence** $s \in S$ must be such that 1. It starts and ends at the first and last text tokens respectively; 2. It uses each text token $\phi_n$ at least once; 3. The sequence can advance by 0 or 1 text token for every advancement of mel-frame.

The likelihood of all valid monotonic alignments is:

$$P(S(\Phi) \mid X) = \sum_{\mathbf{s} \in S(\Phi)} \prod_{t=1}^{T} P(s_t \mid x_t). \qquad (8)$$

which is efficiently implemented using the PyTorch CTC loss module (see Appendix).

**Beta-Binomial Alignment Prior:** We accelerate the alignment learning using a cigar-shaped diagonal prior that promotes the elements on a near-diagonal path formulated using a Beta-Binomial distribution. While formulated differently, it is conceptually similar to the guided attention loss used in (Tachibana et al., 2018), and we believe model training benefits from virtually any reasonable diagonal-shaped prior. It can be visualized in Fig. 2b with details given in the Appendix.

**Extracting $\mathcal{A}_{hard}$:** Alignments generated through duration prediction are discrete. It is therefore necessary to condition the model $g$ on the binarized alignment matrix $\mathcal{A}_{hard}$ to avoid creating a train-test domain gap. We achieve this using the Viterbi algorithm while applying the same constraints for a monotonic alignment as stated above. This gives us the most likely monotonic alignment from the distribution over monotonic paths defined by $\mathcal{A}_{soft}$.

As the Viterbi algorithm is not differentiable, training $g$ conditioned on $\mathcal{A}_{hard}$ would mean that alignment-learning attention mechanism would receive no gradients from $g$. Similar to (Kim et al., 2020), we further enforce that $\mathcal{A}_{soft}$ matches $\mathcal{A}_{hard}$ as much as possible by minimizing their KL-divergence: $\mathcal{L}_{bin} = \mathcal{A}_{hard} \odot \log \mathcal{A}_{soft}$.

### 2.3. Generative Duration Modeling $P_{dur}$

Recent parallel TTS architectures use deterministic regression models to predict durations of lexical units. This limits the amount of diversity achievable during inference in comparison to generative autoregressive models (Valle et al., 2020), where the duration is jointly sampled with other speech characteristics. We address this with a separate normalizing flow model solely for modeling $P_{dur}()$ in Eq. 1. It can be constructed using either another bi-partite flow for full parallelism, or an autoregressive flow similar to the architecture in (Valle et al., 2020).

### 2.4. Training Schedule

Our model uses a training schedule to account for the evolving reliability of extracted alignments. Let $\mathcal{L}_{align}$ be the minimization of the log likelihood of (8). $\mathcal{L}_{mel}$ and $\mathcal{L}_{dur}$ minimize (3) with respect to the decoder flow and phoneme flow respectively. The training begins with the loss function $\mathcal{L} = \mathcal{L}_{mel} + \lambda_1 \mathcal{L}_{align}$, with the corresponding changes applied at given steps:

- [0, 6k): Use $\mathcal{A}_{soft}$ for the alignment matrix.
- [6k, 18k): start using Viterbi $\mathcal{A}_{hard}$ instead of $\mathcal{A}_{soft}$,
- [18k, end): add binarization term $\lambda_2 \mathcal{L}_{bin}$ to the loss.

The duration predictor shares text embeddings with the decoder flow model, but is otherwise fundamentally disjoint and converges rapidly. Thus $\mathcal{L}_{dur}$ is applied once the decoder has converged and its weights frozen.

## 3. Experiments

We investigate the achieved diversity, as well as how the beta-binomial prior influences training stability and alignment quality. For the sake of comparison with prior work, we perform the bulk of our experiments training only on the LJ speech dataset (LJ) (Ito et al., 2017). We use a sampling rate of $22\,050$ Hz and mel-spectrograms with $80$ bins using Librosa's (McFee et al., 2015) mel filter defaults. We apply the STFT with a FFT size of $1024$, window size of $1024$ samples and hop size of $256$ samples ($\sim 12ms$). We use the normalized transcriptions provided in LJ and convert all unambiguous words to phoneme, keeping ambiguous words as graphemes. We use the public WaveGlow model (NVIDIA, 2021) trained on LJ for converting mel-spectrograms to waveforms, setting $\sigma = 0.9$ and denoising strength $0.01$ during inference. For inference from our model, we sample $z_{dur} \sim \mathcal{N}(0, \sigma = .7)$ and $z_{mel} \sim \mathcal{N}_{trunc}(0, \sigma = \{0.5, 0.667\})$, where the latter uses a truncated Normal with truncation at $1.1\sigma$. Mean opinion score and pronunciation error analysis are in the appendix.

### 3.1. Convergence Rate

In order to compare convergence rate, especially early in the training, we turn to the mean mel-cepstral distance (MCD) (Kubichek, 1993). It compares synthesized mel-spectrograms with the ground truth, aligned temporally with dynamic time warping (DTW), and so offers more consistency for cross-model comparisons. While MCD-DTW cannot replace a human subjective quality evaluation of the converged model, it is a reasonable approximation of the audio quality in the early phases of training.

We compare the model with a beta-binomial prior with two baselines: a) a *no-prior* baseline, and b) *boolean* baseline prior obtained by thresholding the betabinomial distribution at $1 \times 10^{-7}$, setting values to 0 below and defining a uniform prior for values above the threshold. This is a strong baseline since it retains cigar shape of the Beta-Binomial. Figure 3 shows MCD-DTW and alignment error. Having any reasonable prior improves both measures, but we note that full beta-binomial prior appears to converge fastest.

### 3.2. Phoneme Duration Distribution

We compare samples and phoneme duration predictions from a deterministic model (Glow-TTS), a test-time dropout model (DP-Dropout-0.5) inspired by its use for output variability in Tacotron2 (Shen et al., 2018), a bi-partite normal-

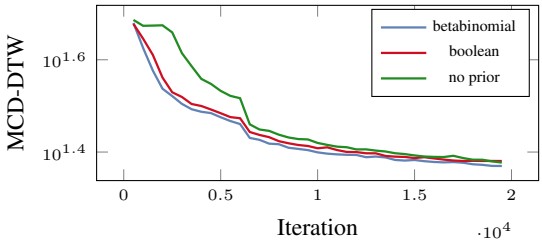

*Figure 3.* The influence of alignment priors during initial 20k steps of training measured by (**top**) DTW-aligned mel-cepstral distortion, and (**bottom**) alignment error w.r.t. 10 manually annotated samples

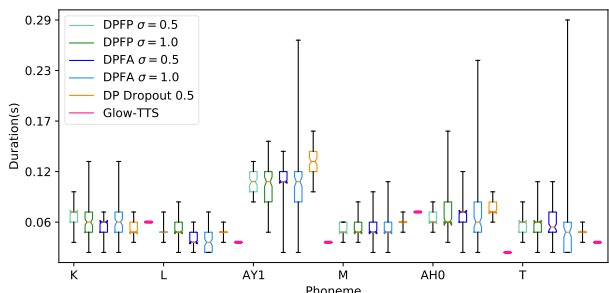

*Figure 4.* Phoneme-level duration distributions for the word *Climate* with $95\%$ confidence intervals obtained from 100 samples collected from different models conditioned on the phrase *Climate change knows no borders*. Explicit generative models (shades of green and blue) provide high diversity in speech rhythm by adjusting $\sigma$, whereas test-time dropout (yellow) provides limited variability.

izing flow (DPFP) and an autoregressive normalizing flow (DPFA). We compute the empirical duration distribution over the word *Climate* in the phrase *Climate change knows no borders* by sampling each model 100 times. We resample whenever the minimum duration is smaller than 25ms and the maximum duration is larger than 300ms.

Figure 4 plots the distribution of phoneme durations, with the proposed architectures producing the most variability. The deterministic baseline cannot synthesize speech with variable prosody. The DP-Dropout-0.5 model produces limited variance and is not as intuitively adjustable as is the $\sigma$ scaling in proper generative modeling.

## 4. Conclusion

We resolve the output diversity issue in parallel TTS architectures by proposing a dedicated generative flow for phoneme duration modeling. Further, our proposed end-to-end alignment architecture extends that of prior works to achieve better convergence rates and better synthesized sample quality as measured on our controlled architecture.

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

# A. Appendix

## A.1. Speech Quality (MOS)

We subjectively compare RAD-TTS samples with Glow-TTS by collecting Mean Opinion Scores (MOS) on utterances from the LJ speech dataset(Ito et al., 2017). Glow-TTS-Blank denotes a variant of Glow-TTS with blank tokens interspersed between all phonemes. RAD-TTS models are trained on LJ with and without the Beta-Binomial attention prior.

We crowd-sourced MOS tests on Amazon Mechanical Turk (Table 1). Raters first pass a sinusoid-counting hearing test for eligibility. They then rate 10 randomly assigned samples from the LJ Test set. Pleasantness and naturalness are rated

on a five-point scale. Over 480 scores were collected per model and each rater was exposed to exactly one model.

Overall, we demonstrate that the inclusion of the prior does improve subjective quality. Compared to the most similar GlowTTS, our overall quality is a bit worse, likely due to our architecture being much larger and therefore less data-efficient on LJSpeech.

| Model | Prior | MOS |
|---|---|---|
| RAD-TTS ($\sigma = .667$) | | $3.40 \pm 0.10$ |
| RAD-TTS ($\sigma = .667$) | ✓ | $3.49 \pm 0.10$ |
| Ground Truth (LJ) | | $4.11 \pm 0.07$ |
| Glow-TTS-Blank ($\sigma = .667$) | | $3.79 \pm 0.09$ |
| Glow-TTS ($\sigma = .667$) | | $3.56 \pm 0.10$ |

*Table 1.* MOS pleasantness and naturalness results along with 95% confidence intervals.

## A.2. Attention Errors

We perform attention error analysis (Peng et al., 2020) on a challenging 100-sentence test set with dates, acronyms, URLs, repeated words, proper nouns, and foreign words in Table 2. Evaluation was performed by manually listening for errors, and may not be directly comparable to previously reported numbers (see bottom bracket in Table 2). As such, we further re-evaluate Glow-TTS at their default $\sigma$-level (0.667), which we find to be similar to our model with both augmentation and prior. Overall, again we see a small improvement incorporating the prior. Our model makes fewer attention/pronunciation errors compared to the original Glow-TTS model, though their blank-interleaved variant performs best by a wide margin.

## A.3. Model architecture

Our bi-partite model is architecture a series of flow steps, each comprising one affine coupling layer and one $1 \times 1$ invertible convolution. **Affine Coupling Layer:** Coupling layers (Dinh et al., 2014; 2016; Kingma & Dhariwal, 2018) are a family of invertible transformations wherein one split of the input data is used to infer scale and translation parameters to affine transform the rest, as described in Algorithm 1.

---

**Algorithm 1** Affine Coupling Layer $\boldsymbol{f}_{coupling}^{-1}(\boldsymbol{x}; context)$

---

**input** $\boldsymbol{x}, context$
**output** $\boldsymbol{s}, \boldsymbol{t}, \boldsymbol{x}'$
  $\boldsymbol{x}_a, \boldsymbol{x}_b = \text{SPLIT}(\boldsymbol{x})$
  $(\boldsymbol{s}, \boldsymbol{t}) = f_{param}(\boldsymbol{x}_a, context)$
  $\boldsymbol{x}'_b = \boldsymbol{s} \odot \boldsymbol{x}_b + \boldsymbol{t}$
  $\boldsymbol{x}' = \text{CONCAT}(\boldsymbol{x}_a, \boldsymbol{x}'_b)$

---

| Model | Prior | Skip | Repeat | Mispron. | Total |
|---|---|---|---|---|---|
| RAD-TTS ($\sigma = .5$) | | 7 | 0 | 6 | 13 |
| RAD-TTS ($\sigma = .5$) | ✓ | 5 | 0 | 7 | 12 |
| Glow-TTS-Blank ($\sigma = .667$) | | 1 | 0 | 5 | 6 |
| Glow-TTS ($\sigma = .667$) | | 4 | 0 | 19 | 23 |
| Glow-TTS ($\sigma = .333$) (Kim et al., 2020) | | 1 | 0 | 3 | 4 |
| ParaNet (Peng et al., 2020) | | 0 | 2 | 4 | 6 |

*Table 2.* Attention error counts on the manually-evaluated Paranet 100-sentence test set. One or more mispronunciations, skips, and repeats count as a single mistake per utterance. Bottom bracket results taken directly from publications and may differ in evaluation criteria.

Here $f_{param}()$ is an arbitrary function that predicts the affine parameters $\boldsymbol{s}, \boldsymbol{b}$ to be applied to the remaining channels $\boldsymbol{x}_b$. In addition to the inputs $\boldsymbol{x}_a$, the parameter prediction is critically conditioned on the *context*, which is a $C_{ctx} \times T$ (same temporal dimension as $X$ as is necessary for parallel processing of each frame) matrix containing combining information for $\Phi, \xi, \mathcal{A}$. The described procedure is fully invertible with respect to the transformation on $\boldsymbol{x}$.

The *context* matrix is setup as follows: The alignment $\mathcal{A}$, visualized in Fig. 2, gives us the number of mel-frames that each $\phi \in \Phi$ should last. We expand the text columns by their corresponding durations by computing $\Phi_{\mathcal{A}} = \Phi\mathcal{A}$, where $\Phi$ and $\mathcal{A}$ are $C_{txt} \times N$ and $N \times T$ respectively. The speaker embedding vector $\xi$ corresponding to the current speaker is then replicated and concatenated to each column. This provides us with a $C_{ctx} \times T$ matrix that temporally aligns text information with the mel-spectrogram frames, as well as providing speaker-dependent information.

**1x1 Invertible Convolution:** As the affine coupling layer always performs the same split, it is necessary to mix up the channels in between applications to ensure every channel is eventually transformed. This can be performed using a fixed permutation matrix (Dinh et al., 2014) to explicitly permute the channel ordering, or to use an invertible, learnable linear transformation $W$ as proposed in (Kingma & Dhariwal, 2018). For our input of 80-dimensional mel spectrograms, $W$ would correspond to a learnable $80 \times 80$ matrix, applied as a kernel-size 1 convolution over $X$. Its formulation and corresponding log determinant are given as:

$$\boldsymbol{f}_{conv}^{-1} = \boldsymbol{W}\boldsymbol{x} \qquad (9)$$

$$\log|\det(\boldsymbol{J}(\boldsymbol{f}_{conv}^{-1}(\boldsymbol{x})))| = \log|\det \boldsymbol{W}| \qquad (10)$$

In practice, we use the LU decomposition trick from (Kingma & Dhariwal, 2018) to accelerate the log determinant computation by storing $\boldsymbol{W}$ as upper and lower triangular matrices. While this significantly accelerates training, it has a minor effect on convergence rates.

## A.4. Compositional Normalizing Flow loss

Taking into the compositional breakdown of $g()$ into $K$ steps, the log-determinant term given in eq. (3) is expanded as:

$$\log\left|\det \mathbf{J}(g^{-1}(x))\right| = \sum_{k=1}^{K} \left(\log\langle|\boldsymbol{s}_k|, \mathbf{1}\rangle + \log|\det \boldsymbol{W}_k|\right)$$

$$(11)$$

## A.5. Beta-Binomial Prior Formulation

We use the beta-binomial distribution which is discrete with a finite support, and probability mass function

$$f_B(k, \alpha, \beta) = \binom{N}{k} \frac{B(k+\alpha)B(N-k+\beta)}{B(\alpha, \beta)} \qquad (12)$$

for $k = \{0, \ldots, N\}$, where $\alpha$ and $\beta$ are hyperparameters of the beta function $B(\cdot, \cdot)$. We use the beta-binomial distribution to construct a 2D cigar-shaped prior over the diagonal of $\mathcal{A}_{soft}$, which widens in the center of the matrix, and narrows towards the corners. For every column of $\mathcal{A}_{soft}$ representing $P(\Phi \mid X{=}x_t)$, we incorporate the prior to attain the posterior:

$$P_{posterior}(\Phi{=}\phi_k \mid X{=}x_t) =$$
$$P(\Phi{=}\phi_k \mid X{=}x_t) \odot f_B(k, \omega t, \omega(T-t+1)) \quad (13)$$

for $k = \{0, \ldots, N\}$, where $\omega$ is the scaling factor controlling the width: lower the $\omega$, wider is the width of the prior.

## A.6. CTC Formulation

**Generic HMM Formulation:** Let $\lambda = (A, B, \pi)$ be the parameters of an HMM: $A$ the state transition matrix, $B$ the emission probability matrix, and $\pi$ the vector of initial state probabilities. Denote by $b_i(j)$ the emission probability of symbol $j$ in the $i$th state.

For an observation sequence $O = \{O_1, \ldots, O_n\}$, the probability of such sequence can be computed recursively (Rabiner, 1990)

$$\alpha_1(i) = \pi_i b_i(O_i) \quad 1 \le i \le N, \qquad (14)$$

$$\alpha_{t+1}(j) = \left[\sum_{j=1}^{N} \alpha_t(i) a_{ij}\right] b_j(O_{t+1}) \qquad (15)$$

$$\text{for } 1 \le t \le T-1, \quad 1 \le j \le N.$$

The final probability is the sum of final forward variables over all states

$$P(O \mid \lambda) = \sum_{i=1}^{N} \alpha_T(i). \qquad (16)$$

CTC imposes several constraints on this general formulation. Let

```
The quick brown fox
```

be a sample transcription of an input audio signal. In CTC, it is being interleaved with blank tokens $\emptyset$, giving raise to the sequence of observations

$$O = \{\emptyset, \texttt{T}, \emptyset, \texttt{h}, \emptyset, \texttt{e}, \ldots, \texttt{f}, \emptyset, \texttt{o}, \emptyset, \texttt{x}, \emptyset).$$

In CTC, initial state probabilities $\pi$ are not modeled. The state-transition matrix $A$ is sparse and allows only a handful of monotonic transitions from neighboring states, giving raise to the recursive formulation of Eq. (15) in CTC

$$\alpha_t(s) = \begin{cases} [\alpha_{t-1}(s) + \alpha_{t-1}(s-2)]\, b_t(O_s) \\ \quad \text{if } O_s = \emptyset \text{ or } O_s = O_{s-2}, \\ \\ [\alpha_{t-1}(s) + \alpha_{t-1}(s-2) + \alpha_{t-1}(s-1)]\, b_t(O_s) \\ \quad \text{otherwise .} \end{cases}$$

$$(17)$$

It tells that a transition can occur only to the same ($\alpha_{t-1}(s)$), or a subsequent state ($\alpha_{t-1}(s-1)$), and blank states can be skipped altogether ($\alpha_{t-1}(s{-}2)$). The condition $O_s = O_{s-2}$ ensures that when there are repeated letters (e.g., in the word Anna), the blank state has to be visited in order to later differentiate between both occurrences of the repeated letter.

When using CTC, the speech recognition model outputs an emission probability matrix $B \in \mathbb{R}^{|V|\times T}$, where $|V|$ denotes the size of the vocabulary with the blank symbol (and so the number of states), and $T$ is the number of outputs proportional to the number of input audio frames. Every column in $B$ is normalized to a probability distribution.

**Calculating $\mathcal{A}_{soft}$:** In RAD-TTS, the alignment matrix $\mathcal{A}_{soft} \in \mathbb{R}^{N\times T}$ can be interpreted similarly to $B$, as a series of column-wise emission probability distributions over possible $N$ input symbol (states) for each of $T$ audio frames. To ensure monotonicity of alignment, we build a surrogate transcript with a sequence of unique states

$$\{s_1, s_2, \ldots, s_N\}$$

which, upon passing to CTC loss, is being converted to

$$O = \{\emptyset, s_1, \emptyset, s_2, \ldots, \emptyset, s_N, \emptyset\}.$$

As in CTC, we do not model initial probabilities $\pi$. In order to account for the blank symbol state, we add a row to $\mathcal{A}_{soft}$

$$\begin{bmatrix} \mathcal{A}_{soft} \\ \epsilon \end{bmatrix},$$

where $\epsilon$ denotes a constant blank probability for every frame. We set to a tiny, non-zero value, to prevent numerical instability. The recursive formulation from Eq. (17) can now be broken down into two cases:

1. $O_t = \emptyset$ (odd positions in the transcript denoted $2k+1$).

   From $b_t(\emptyset) = 0$ it follows that $\alpha_t(2k+1) = 0$.

2. $O_t \neq \emptyset$ (even positions in the transcript denoted $2k$).

   The surrogate sequence has unique states and $O_t \neq \emptyset$. It follows that $O_t \neq O_{t-2}$, and the first case of Eq. (17) is not possible. Because $\alpha_{t-1}(2k-1) = 0$, we have

   $$\alpha_t(2k) = [\alpha_{t-1}(2k) + \alpha_{t-1}(2k-2)] \; b_t(O_s).$$

Thus, with negligible blank probability and under the imposed monotonicity constraint, Eq. (17) effectively reduces to Eq. (15) from the original formulation for calculating forward variables.

In practice, setting the blank emission probability $\epsilon$ to be roughly the value of the largest of the initial activations significantly improves convergence rates. The reasoning behind this is that it relaxes the monotonic constraint, allowing the objective function to construct paths while optionally skipping over some text tokens, notably ones that have not been sufficiently trained on during early iterations. As training proceeds, the probabilities of the skipped text token increases, despite the existence of the blank tokens, allowing us to extract clean monotonic alignments.

**Example Implementation:**

```python
class ForwardSumLoss(torch.nn.Module):
    def __init__(self, blank_logprob=-1):
        super(ForwardSumLoss, self).__init__()
        self.log_softmax = torch.nn.LogSoftmax(dim=3)
        self.blank_logprob = blank_logprob
        self.CTCLoss = nn.CTCLoss(zero_infinity=True)

    def forward(self, attn_logprob, text_lens, mel_lens):
        """
        Args:
            attn_logprob: batch x 1 x max(mel_lens) x max(text_lens)
                    batched tensor of attention log
                    probabilities, padded to length
                    of longest sequence in each dimension
            text_lens: batch-D vector of length of
                    each text sequence
            mel_lens: batch-D vector of length of
                    each mel sequence

        """
        # The CTC loss module assumes the existence of a blank token
        # that can be optionally inserted anywhere in the sequence for
        # a fixed probability.
        # A row must be added to the attention matrix to account for this
        attn_logprob_pd = F.pad(input=attn_logprob,
                                pad=(1, 0, 0, 0, 0, 0, 0, 0),
                                value=self.blank_logprob)
        cost_total = 0.0
        # for-loop over batch because of variable-length
        # sequences
        for bid in range(attn_logprob.shape[0]):
            # construct the target sequence. Every
            # text token is mapped to a unique sequence number,
            # thereby ensuring the monotonicity constraint
            target_seq = torch.arange(1, text_lens[bid]+1)
            target_seq=target_seq.unsqueeze(0)

            curr_logprob = attn_logprob_pd[bid].permute(1, 0, 2)
            curr_log_prob = curr_logprob[:mel_lens[bid],:,:text_lens[bid]+1]
            curr_logprob = self.log_softmax(curr_logprob[None])[0]
            cost = self.CTCLoss(curr_logprob,
                                target_seq,
                                input_lengths=mel_lens[bid:bid+1],
                                target_lengths=text_lens[bid:bid+1])
            cost_total += cost
        # average cost over batch
        cost_total = cost_total/attn_logprob.shape[0]
        return cost_total
```