# OpenReview forum: "RAD-TTS: Parallel Flow-Based TTS with Robust Alignment Learning and Diverse Synthesis"
_ICML.cc/2021/Workshop/INNF — INNF+ 2021 poster_

### Official Review · Reviewer_wUxY · 2021-06-10

**Rating:** Borderline Reject
**Confidence:** 3

**Summary:**

The authors propose a generative model for text-to-speech synthesis based on flows. Different from prior works, the proposed model also treats phoneme-duration as a random variable and models it using a normalizing flow. As a consequence, their model generates more diverse samples. The authors also propose a specific prior on the alignment matrix to speed up inference.

**Justification For Rating:**

Indeed, it appears that the proposed model increases sample variability. However (from Figure 1), it seems like the model with our prior slightly increases MOS but can only outperform GLOW-TTS using additional data. I believe a fair comparison should also include GLOW-TTS trained using the additional data. The authors also do not address how their model is more stable nor show evidence that the prior speeds up training.



minor observations:
* The TTS acronym is never introduced
* line 37, column 1: 'these concern' -> 'this concern' or 'these concerns'
* line 2, column 2: 'Most related work' -> 'The most ...' -- additionally, it is difficult to appreciate this sentence (and the next) here
	, without knowing what you do in more detail.
* line 96: 'We wish to...' seems imprecise. I guess the objective is to model x as a bijective transformation of z, the latter being standard Normal.
* line 56, column 2: 'Inference' is a very broad word (encompasses, e.g., parameter estimation and taking intervals), 'sampling' would be more precise.

---

### Official Review · Reviewer_Khg3 · 2021-06-11

**Rating:** Borderline Accept
**Confidence:** 2

**Summary:**

The paper presents a flow-based text-to-speech synthesis algorithm that uses a generative model only for token durations (speech rhythm). The main goal is to increase the diversity. The model consists of two flows: a phoneme flow that generates phoneme durations, and a MEL flow that generates MEL-spectogram. The proposed model seems to yield larger diversity in phoneme duration. The pleasantness seems to be on par with Glow-TTS variants.

**Justification For Rating:**

The paper does not seem to present a big departure from existing approaches. The results seems somewhat on par, although the diversity in phoneme duration is measurably higher than that of prior approaches. That appears to be a good accomplishment, but I'm not an expert in this area and cannot therefore assess the importance of the technical contribution. Other than that, the paper is well written and could trigger follow-up discussions in the TTS community. I'm thus not opposed to acceptance.

---

### Decision · Program_Chairs · 2021-06-15

**Decision:**

Accept (poster)

**Comment:**

The paper is on topic for the workshop. The reviews are mixed, with the main positive point being that the sample diversity is increased, as is one of the claims made in the paper. The main points of criticism are: 1) that the proposed method is only a minor deviation of existing approaches, 2) that the model can only outperform the baseline model in terms of MOS scores when using additional data for the proposed method while not allowing the baseline method to do so, and finally, 3) that the claims of increased stability and speed up in training are not backed up by evidence. We have decided to accept this paper, but urge the authors  to take into account the the reviewer's comments for the camera ready version, especially point 2) and 3).